# Subgraph Neural Networks

**Emily Alsentzer**[*]
Harvard University, MIT
emilya@mit.edu

**Samuel G. Finlayson**[*]
Harvard University, MIT
sgfin@mit.edu

**Michelle M. Li**
Harvard University
michelleli@g.harvard.edu

**Marinka Zitnik**
Harvard University
marinka@hms.harvard.edu

## Abstract

Deep learning methods for graphs achieve remarkable performance on many node-level and graph-level prediction tasks. However, despite the proliferation of the methods and their success, prevailing Graph Neural Networks (GNNs) neglect subgraphs, rendering subgraph prediction tasks challenging to tackle in many impactful applications. Further, subgraph prediction tasks present several unique challenges: subgraphs can have non-trivial internal topology, but also carry a notion of position and external connectivity information relative to the underlying graph in which they exist. Here, we introduce SUBGNN, a subgraph neural network to learn disentangled subgraph representations. We propose a novel subgraph routing mechanism that propagates neural messages between the subgraph's components and randomly sampled anchor patches from the underlying graph, yielding highly accurate subgraph representations. SUBGNN specifies three channels, each designed to capture a distinct aspect of subgraph topology, and we provide empirical evidence that the channels encode their intended properties. We design a series of new synthetic and real-world subgraph datasets. Empirical results for subgraph classification on eight datasets show that SUBGNN achieves considerable performance gains, outperforming strong baseline methods, including node-level and graph-level methods, by $19.8\%$ over the strongest baseline. SUBGNN performs exceptionally well on challenging biomedical datasets where subgraphs have complex topology and even comprise multiple disconnected components.

## 1 Introduction

Deep learning on graphs and Graph Neural Networks (GNNs), in particular, have emerged as the dominant paradigm for learning compact representations of interconnected data [66, 81, 23]. The methods condense graph neighborhood connectivity patterns into low-dimensional embeddings that can be used for a variety of downstream prediction tasks. While graph representation learning has made tremendous progress in recent years [20, 84], prevailing methods focus on learning useful representations for nodes [25, 68], edges [21, 37] or entire graphs [6, 27].

Graph-level representations provide an overarching view of the graphs but at the loss of some finer local structure. In contrast, node-level representations focus instead on the preservation of the local topological structure (potentially to the detriment of the big picture). It is unclear if the methods can generate powerful representations for subgraphs and effectively capture the unique topology of subgraphs. Despite the popularity and importance of subgraphs for machine learning [77, 13, 78], there is still limited research on subgraph prediction [41], *i.e.*, to predict if a particular subgraph has

---

[*]Contributed Equally.

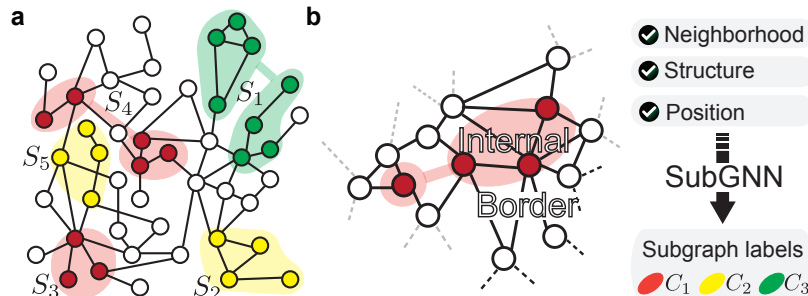

Figure 1: **a.** Shown is graph $G$ in which nodes explicitly state their group memberships, resulting in intricate subgraph structures, $\mathcal{S} = \{S_1, \ldots, S_5\}$. Subgraphs $S_2$, $S_3$ and $S_5$ comprise single connected components in $G$ whereas subgraphs $S_1$ and $S_4$ each form two isolated components. Colors indicate subgraph labels, $\mathcal{C} = \{C_1, C_2, C_3\}$. **b.** We investigate the problem of predicting subgraph properties $\mathcal{C}$ by learning subgraph representations that recognize and disentangle the heterogeneous properties of subgraphs (*i.e.*, neighborhood, structure, and position) and how they relate to underlying $G$ (*i.e.*, internal connectivity and border structure of the edge volume that points outside of the subgraph).

a particular property of interest (Figure 1a). This can be, in part, because subgraphs are incredibly challenging structures from the topological standpoint (Figure 1b). (1) Subgraphs require that we make joint predictions over larger structures of varying sizes—the challenge is how to represent subgraphs that do not coincide with enclosing $k$-hop neighborhoods of nodes and can even comprise multiple disparate components that are far apart from each other in the graph. (2) Subgraphs contain rich higher-order connectivity patterns, both internally among member nodes as well as externally through interactions between the member nodes and the rest of the graph—the challenge is how to inject information about border and external subgraph structure into the GNN's neural message passing. (3) Subgraphs can be localized and reside in one region of the graph or can be distributed across multiple local neighborhoods—the challenge is how to effectively learn about subgraph positions within the underlying graph in which they are defined. (4) Finally, subgraph datasets give rise to unavoidable dependencies that emerge from subgraphs sharing edges and non-edges—the challenge is how to incorporate these dependencies into the model architecture while still being able to take feature information into account and facilitate inductive reasoning.

**Present work.** Here, we introduce SUBGNN[2] (Figure 2), a novel graph neural network for subgraph prediction that addresses all of the challenges above. Unlike current prediction problems that are defined on individual nodes, pairwise relations, or entire graphs, our task here is to make predictions for subgraphs. To the best of our knowledge, SUBGNN is the only representation learning method designed for general subgraph classification. While [41] performs prediction on small (3-4 nodes), fixed-size subgraphs, SUBGNN operates on larger subgraphs with varying sizes and multiple disparate components. SUBGNN's core principle is to propagate messages at the subgraph level via three property-aware channels that capture position, neighborhood, and structure. Further, SUBGNN is inductive: it can operate on new subgraphs in unseen portions of the graph.

Experiments on eight datasets show that SUBGNN outperforms baselines by an average of 77.4% on synthetic datasets and 125.2% on real-world datasets. Further, SUBGNN improves the strongest baseline by 19.8%. As an example, on a network of phenotypes where subgraphs represent symptoms of a disease, our method is able to distinguish between 10 subcategories of monogenic neurological disorders. We also find that several naive generalizations of node-level and graph-level GNNs have poor performance. This finding is especially relevant as these generalizations are popular in practical applications, yet they cannot fully realize the potential of neural message-passing for subgraph prediction, as evidenced by our experiments. For example, we find that a popular node-level approach, which represents subgraphs with virtual nodes (*i.e.*, meta nodes) and then uses an attention-based GNN (*e.g.*, GAT [62]) to train a node classifier, performs poorly in a variety of settings.

Finally, we design a suite of synthetic subgraph benchmarks that are uniquely suited for evaluating aspects of subgraph topology. We also provide four new and original real-world datasets from

biological, medical, and social domains to practically test the new algorithms. Each dataset consists of an underlying graph overlaid with a large number of subgraphs whose labels we need to predict.

## 2 Related Work

We proceed by reviewing major threads of research relevant for subgraph representation learning.

**Subgraph embeddings and prediction.** A few recent works learn representations of small, localized subgraphs. [10] encodes the subgraph of entities connected to the candidate answer entity in a knowledge graph for question answering, and [41] encodes 3-node or 4-node subgraphs for subgraph evolution prediction. In contrast, SUBGNN can learn representations for large, variable-size subgraphs that can be distributed throughout the graph.

**Subgraph extraction and network community detection.** The topology of border structure has been extensively examined in the context of community detection [71, 46, 67] (also known as module detection and graph clustering), highlighting its importance for network science [8, 47]. However, community detection, motif counting, and embedding methods for subgraph extraction and kernel-based similarity estimation [53, 58, 74, 63, 30, 15, 2, 35] are fundamentally different from SUBGNN. These methods search for groups of nodes that are well-connected internally while being relatively well-separated from the rest of the graph and typically limited to individual connected components. In contrast, SUBGNN is a subgraph prediction framework that reasons over a given set of subgraphs.

**Learning representations of higher-order structures, ego nets, and enclosing subgraphs.** Hypergraph neural networks [82] and their variants [54, 18, 79, 45, 80] have become a popular approach for learning on hypergraph-structured data. These methods use the spectral theory of hypergraphs [70] or clique expansion [1, 29, 31] to formulate higher-order message passing, where nodes receive latent representations from their immediate (one-hop) neighbors and from further $N$-hop neighbors at every message passing step. Recent studies, *e.g.*, [76, 44, 64, 10], also learn representations for permutation-invariant functions of sequences (or multiset functions [40]). These methods typically express permutation-invariant functions as the average of permutation-sensitive functions applied to all reorderings of the input sequence. While this approach allows for pooling operators that are invariant to the ordering of nodes, it does not account for rich subgraph dependencies, such as when subgraphs share edges and have many connections to the rest of the graph.

**Subgraphs and patches in GNNs.** Subgraph structure is key for numerous graph-related tasks [61, 16, 73]. For example, Patchy-San [49] uses local receptive fields to extract useful features from graphs. Ego-CNN uses ego graphs to find critical structures [60]. SEAL [78] develops a theory showing that enclosing subgraphs are sufficient for link prediction. Building on this evidence, GraIL [59] extracts local subgraphs induced by nodes occurring on the shortest paths between the target nodes and uses them for inducing logical rules. Cluster-GCN [13], Ripple walks [5], and GraphSAINT [77] use subgraphs to design more efficient and scalable algorithms for training deep and large GNNs. While these methods use substructures to make GNN architectures more efficient or improve performance on node and edge prediction tasks, none of the methods consider prediction on subgraphs.

## 3 Formulating Subgraph Prediction

Let $G = (V, E)$ denote a graph comprised of a set of edges $E$ and nodes $V$. While we focus on undirected graphs, it is straightforward to extend this work to directed graphs. $S = (V', E')$ is a subgraph of $G$ if $V' \subseteq V$ and $E' \subseteq E$. Each subgraph $S$ has a label $y_S$ and may consist of multiple connected components, $S^{(C)}$, which are defined as a set of nodes in $S$ such that each pair of nodes is connected by a path (*e.g.*, see Figure 1). Importantly, the number and size of the subgraphs are bounded by, but do not directly depend on, the number of nodes in the graph $G$.

**Background on Graph Neural Networks.** Many graph neural networks, including ours, can be formulated as message passing networks (MPN) [7, 65, 83]. Message-passing networks are defined by three functions, MSG, AGG, and UPDATE, which are used to propagate signals between elements of the network and update their respective embeddings. Typically, these functions are defined to operate at the node level, propagating messages to a node $v_i$ from the nodes in its neighborhood $\mathcal{N}_{v_i}$. In this typical context, a message between a pair of nodes $(v_i, v_j)$ at layer $l$ is defined as a function of the hidden representations of the nodes $\mathbf{h}_i^{l-1}$ and $\mathbf{h}_j^{l-1}$ from the previous layer: $m_{ij}^l = \text{MSG}(\mathbf{h}_i^{l-1}, \mathbf{h}_j^{l-1})$.

In AGG, messages from $\mathcal{N}_{v_i}$ are aggregated and combined with $\mathbf{h}_i^{l-1}$ to produce $v_i$'s representation for layer $l$ in UPDATE. Many extensions of the canonical message passing framework have been proposed [7, 62, 13, 22], *e.g.*, [75] passes messages from a shared set of anchor nodes rather than a strict neighborhood, to allow node embeddings to capture global position.

### 3.1  SUBGNN: Problem Formulation

**Problem (Subgraph Representations and Property Prediction).** *Given subgraphs* $\mathcal{S} = \{S_1, S_2, \ldots, S_n\}$, SUBGNN *specifies a neural message passing architecture* $E_S$ *that generates a* $d_s$-*dimensional subgraph representation* $\mathbf{z}_S \in \mathbb{R}^{d_s}$ *for every subgraph* $S \in \mathcal{S}$. SUBGNN *uses the representations to learn a subgraph classifier* $f : \mathcal{S} \to \{1, 2, \ldots, C\}$ *for subgraph labels* $f(S) = \hat{y}_S$.

Note that while this paper focuses on the subgraph classification task, the methods we propose entail learning an embedding function $E_S : S \to \mathbb{R}^{d_s}$ that maps each subgraph to a low-dimensional representation that captures the key aspects of subgraph topology necessary for prediction. All the techniques that we introduce to learn $E_S$ can be extended without loss of generality to other supervised, unsupervised, or self-supervised tasks involving subgraphs.

In this work, we define message passing functions that operate at the *subgraph* level. This allows us to explicitly capture aspects of subgraph representation that do not apply to nodes or whole graphs. In particular, messages are propagated to each connected component in the subgraph, allowing us to build meaningful representations of subgraphs with multiple distinct connected components.

**SUBGNN: Properties of subgraph topology.** Subgraph representation learning requires models to encode network properties that are not necessarily defined for either nodes or graphs. Subgraphs have non-trivial internal structure, border connectivity, and notions of neighborhood and position relative to the rest of the graph. Intuitively, our goal is to learn a representation of each subgraph $S_i$ such that the likelihood of preserving certain properties of the subgraph is maximized in the embedding space. Here we provide a framework of six properties of subgraph structure that are key for learning powerful subgraph representations (Table 1).

Table 1: Six properties of subgraph topology in SUBGNN. Internal nodes correspond to nodes within the subgraph $S_i$ in graph $G$, and border nodes correspond to nodes within the $k$-hop neighborhood of any node in $S_i$. See also Figure 1.

| SUBGNN Channel | SUBGNN Subchannel | |
| --- | --- | --- |
| | Internal (I) | Border (B) |
| Position (P) | Distance between $S_i$'s components | Distance between $S_i$ and rest of $G$ |
| Neighborhood (N) | Identity of $S_i$'s internal nodes | Identity of $S_i$'s border nodes |
| Structure (S) | Internal connectivity of $S_i$ | Border connectivity of $S_i$ |

**1) Position.** Two notions of position may be defined for subgraphs. The first, *border position*, refers to its distance to the rest of $G$; this is directly analogous to node position, which can distinguish nodes with isomorphic neighborhoods [75]. The second, *internal position*, captures the relative distance between components within the subgraph.

**2) Neighborhood.** Subgraphs extend the notion of a node's neighborhood to include both internal and external elements. *Border neighborhood*, as with nodes, is defined as the set of nodes within $k$ hops of any node in $S$. Each connected component in a subgraph will have its own border neighborhood. Subgraphs also have a non-trivial *internal neighborhood*, which can vary in size and position.

**3) Structure.** The connectivity of subgraphs is also non-trivial. A subgraph's *internal structure* is defined as the internal connectivity of each component. Simultaneously, subgraphs have a *border structure*, defined by edges connecting internal nodes to the border neighborhood.

We expect that capturing each of these properties will be crucial for learning useful subgraph representations, with the relative importance of these properties determined by the downstream task.

## 4  SUBGNN: SUBGRAPH NEURAL NETWORK

Next, we describe SUBGNN, our approach to subgraph classification, which is designed to learn representations that can capture the six aspects of subgraph structure from Table 1. SUBGNN

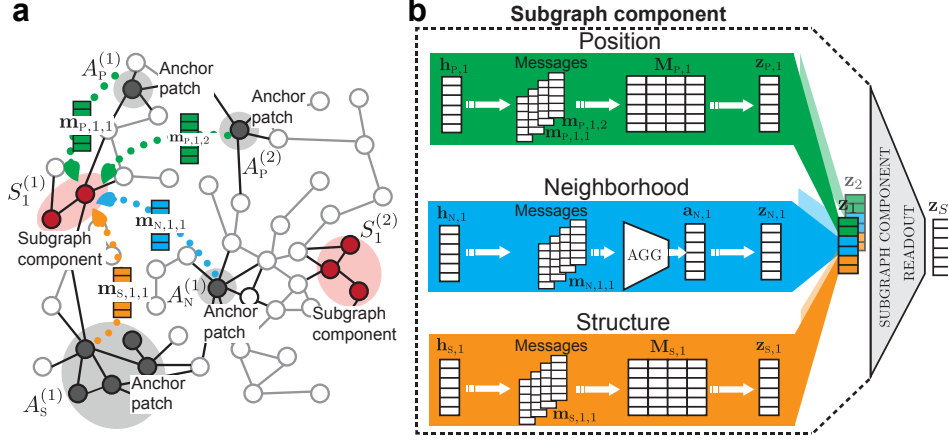

Figure 2: SUBGNN architecture. **a.** Property-specific messages $\text{MSG}_{\text{X}}^{A \to S}$ are propagated from anchor patches $A_{\text{X}}$ to components of subgraph $S$. Here, $\mathbf{m}_{\text{X},c,i}$ denotes property-X message from $i$-th anchor patch $A_{\text{X}}^{(i)}$ to $c$-th component $S^{(c)}$. **b.** SUBGNN specifies three channels, which are each designed to capture a distinct subgraph property. Channel outputs $\mathbf{z}_{\text{X}}$ are concatenated to produce a final subgraph representation $\mathbf{z}_S$.

learns representations of subgraphs in a hierarchical fashion, by propagating neural messages from anchor patches to subgraph components and aggregating the resultant representations into a final subgraph embedding (Figure 2a). Message-passing for each connected component takes place in independent position, neighborhood, and structure channels, each explicitly optimized to encode a distinct subgraph property (Figure 2b). The messages are sent from anchor patches sampled from throughout the graph and weighted by a property-specific similarity function. This approach is modular, extensible, and can be customized based on knowledge of the downstream task.

## 4.1 Subgraph-Level Message Passing

The core of our approach is a message passing framework, as described in Table 1. SUBGNN specifies a novel aggregation scheme that is defined at the level of subgraph components. SUBGNN specifies how to propagate neural messages from a set of anchor patches to subgraph components and, eventually, to entire subgraphs, resulting in subgraph representations that capture distinct properties of subgraph topology (Table 1).

Anchor patches $\mathcal{A}_{\text{X}} = \{A_{\text{X}}^{(1)}, \dots, A_{\text{X}}^{(n_A)}\}$ are subgraphs, which we randomly sample from $G$ in a channel-specific manner, resulting in anchor patches $\mathcal{A}_{\text{P}}$, $\mathcal{A}_{\text{N}}$, and $\mathcal{A}_{\text{S}}$ for each of three SUBGNN's channels, position, structure, and neighborhood, respectively. We define the message from anchor patch $A_{\text{X}}$ to subgraph component $S^{(c)}$ as follows:

$$\text{MSG}_{\text{X}}^{A \to S} = \gamma_{\text{X}}(S^{(c)}, A_{\text{X}}) \cdot \mathbf{a}_{\text{X}} \tag{1}$$

where X is the channel, $\gamma_{\text{X}}$ a similarity function between the component $S^{(c)}$ and the anchor patch $A_{\text{X}}$, and $\mathbf{a}_{\text{X}}$ is the learned representation of $A_{\text{X}}$. Similarity function $\gamma_{\text{X}}$ is any learned or pre-specified similarity measure quantifying the relevance of an anchor patch to a subgraph. These messages are then transformed into order-invariant hidden representation $\mathbf{h}_{\text{X},c}$ and subgraph component $S^{(c)}$ as:

$$\begin{aligned} \mathbf{g}_{\text{X},c} &= \text{AGG}_M(\{\text{MSG}_{\text{X}}^{A_{\text{X}} \to S^{(c)}} \; \forall A_{\text{X}} \in \mathcal{A}_{\text{X}}\}), \\ \mathbf{h}_{\text{X},c} &\leftarrow \sigma(\mathbf{W}_{\text{X}} \cdot [\mathbf{g}_{\text{X},c}; \mathbf{h}_{\text{X},c}]), \end{aligned} \tag{2}$$

where $\mathbf{W}_{\text{X}}$ is a layer-wise learnable weight matrix for channel X, $\sigma$ is a non-linear activation function, $\text{AGG}_M$ is an aggregation function that operates over messages, and $\mathbf{h}_{\text{X},c}$ is the representation at the previous layer that gets updated. Note that $c$ is shorthand notation for $S^{(c)}$. Eq. (2) outputs a channel-specific hidden representation $\mathbf{h}_{\text{X},c}$ for component $S^{(c)}$ and channel X. This hidden representation is then passed to the next layer of the SUBGNN architecture.

The order invariance of $\mathbf{h}_{\mathrm{x},c}$ is a necessary property for layer-to-layer message passing; however, it limits the ability to capture the structure and position of subgraphs [75]. Because of that, we create property-aware output representations, $\mathbf{z}_{\mathrm{x},c}$, by producing a matrix of anchor-set messages $\mathbf{M}_{\mathrm{x}}$, where each row is an anchor-set message computed by $\mathrm{MSG}_{\mathrm{x}}$, and then passing that matrix through a non-linear activation function (Figure 2b and Algorithm 1). In the resulting representation, each dimension of the embedding encodes the structural or positional message from anchor patch $A$. For the neighborhood channel, we set $\mathbf{z}_{\mathrm{N},c} = \mathbf{h}_{\mathrm{N},c}$. Finally, SUBGNN routes messages for internal and border properties within subchannels for each channel P, N and S, (*i.e.*, $\{\mathrm{P_I}, \mathrm{P_B}\}, \{\mathrm{N_I}, \mathrm{N_B}\}, \{\mathrm{S_I}, \mathrm{S_B}\}$) and concatenates the final outputs.

Property-aware output representations $\mathbf{z}_{\mathrm{x},c}$ are transformed into final subgraph representations through order-invariant functions defined at the channel-, layer-, and component-level. Channel-specific representations $\mathbf{z}_{\mathrm{x},c}$ are first aggregated into a subgraph component representation for each layer, via channel aggregation function $\mathrm{AGG}_C$. We then aggregate component representations across all layers via $\mathrm{AGG}_L$ to produce a final subgraph component representation $\mathbf{z}_c$ [69]. Finally, the component representations $\mathbf{z}_c$ for every component $S^{(c)}$ are aggregated into a final subgraph representation $\mathbf{z}_S$ via READOUT. A full description of SUBGNN is detailed in Appendix A.

## 4.2 Property-Aware Routing

We enforce property-awareness through the creation of dedicated routing channels for position, neighborhood, and structure. Each channel X has three key elements: (1) a sampling function $\phi_{\mathrm{X}}$ to generate anchor patches, (2) an anchor patch encoder $\psi_{\mathrm{X}}$, and (3) a similarity function $\gamma_{\mathrm{X}}$ to weight messages sent between anchor patches and subgraph components.

**(1) Sampling anchor patches.** For each subchannel, we define an anchor patch sampling function $\phi_{\mathrm{X}} : (G, S^{(c)}) \rightarrow A_{\mathrm{X}}$. In the position channel, the internal anchor patch sampler $\phi_{\mathrm{P_I}}$ returns patch $A_{\mathrm{P_I}}$ comprised of a single node sampled from subgraph $S$. The resulting set of anchor patches $\mathcal{A}_{\mathrm{P_I}}$ is shared across all components in $S$. This allows representations of $S$'s components to be positioned relative to each other. In contrast, the border position anchor patch sampler is designed to position subgraphs relative to the rest of the graph. Accordingly, $\phi_{\mathrm{P_B}}$ samples nodes such that anchor patches $\mathcal{A}_{\mathrm{P_B}}$ are shared across all subgraphs. In the neighborhood channel, the internal anchor patch sampler $\phi_{\mathrm{N_I}}$ samples nodes from the subgraph component $S^{(c)}$, and the border patch sampler $\phi_{\mathrm{N_B}}$ samples nodes from the $k$-hop neighborhood of $S^{(c)}$. The structure anchor patch sampler $\phi_{\mathrm{S}}$ is used for both internal and border structure, and the resulting $\mathcal{A}_{\mathrm{S_I}}$ and $\mathcal{A}_{\mathrm{S_B}}$ are shared across all $S$. In particular, $\phi_{\mathrm{S}}$ returns a connected component sampled from the graph via triangular random walks, *i.e.*, second-order, biased random walks that extend classical random walks to privilege the traversal of triangles [9] and allow for effective capture of structural properties in graphs [24, 9, 11]. Refer to Appendix A.2 for details.

**(2) Neural encoding of anchor patches.** Each channel X specifies an anchor patch encoder $\psi_{\mathrm{X}} : A_{\mathrm{X}} \rightarrow \mathbf{a}_{\mathrm{X}}$ that encodes anchor patch $A_{\mathrm{X}}$ into a $d$-dimensional embedding $\mathbf{a}_{\mathrm{X}}$. Note that $\psi_{\mathrm{N}}$ and $\psi_{\mathrm{P}}$ simply map to the node embedding of the anchor patch node. In contrast, $\psi_{\mathrm{S}}$ returns representations of structure anchor patches. To generate representations for the structure anchor patches, we first perform $w$ fixed-length triangular random walks with parameter $\beta$ to produce a sequence of traversed nodes $(u_{\pi_w(1)}, \ldots, u_{\pi_w(n)})$ (See Appendix A.2). The resulting sequence of node embeddings is then fed into a bidirectional LSTM, whose hidden states are summed to produce a final representation, $\mathbf{a}_{\mathrm{S}}$. To capture the internal structure of the anchor patch, we perform random walks within the anchor patch $\mathcal{I} = \{u | u \in A_{\mathrm{S}}\}$. Separately, to capture the border structure of the anchor patch, we perform random walks over the external neighborhood $\mathcal{E} = \{u | u \notin A_{\mathrm{S}}\}$ and the border nodes $\mathcal{B} = \{u | u \in \mathcal{I}, v \in \mathcal{E}, e_{uv} \in E\}$. We limit $\mathcal{E}$ to external nodes within $k$ hops of any node in $A_{\mathrm{S}}$.

**(3) Estimating similarity of anchor patches.** The similarity function $\gamma_{\mathrm{X}} : (S^{(c)}, A_{\mathrm{X}}) \rightarrow [0, 1]$ determines the relative weighting of each anchor patch in building the subgraph component representations. In principle, $\gamma_{\mathrm{X}}$ can be any learned or predefined similarity function. For the position channel, we define $\gamma_{\mathrm{P}}$ as a function of the shortest path (SP) on the graph between the connected component $S^{(c)}$ and the anchor patch $A_{\mathrm{P}}$, $\gamma_{\mathrm{P}}(S^{(c)}, A_{\mathrm{P}}) = 1/(d_{sp}(S^{(c)}, A_{\mathrm{P}}) + 1)$, where $d_{sp}$ represents the average SP length. For the neighborhood channel, $\gamma_{\mathrm{N}}(S^{(c)}, A_{\mathrm{N}})$ is defined similarly, though note that, due to the sampling schemes, this evaluates to $\gamma_{\mathrm{N_I}} = 1$ in the case of internal neighborhood and $\gamma_{\mathrm{N_B}} \leq k$ in context of the $k$-hop border neighborhood sampler. To measure subgraph structure in a permutation-

invariant manner, we compute a canonical input ordering [44], defined by ordered degree sequence, for the component $S^{(c)}$ and anchor patch $A_s$. These are compared using the normalized dynamic time warping (DTW) measure [43]. Specifically, we define $\gamma_s(S^{(c)}, A_s) = 1/(\text{DTW}(d_{S^{(c)}}, d_{A_s}) + 1)$, where $d_{S^{(c)}}$ and $d_{A_s}$ are the ordered degree sequences for the subgraph component and anchor patch, respectively. The degree for each node $u \in S^{(c)}$ or $A_s$ is defined for internal structure as the number of edges from $u$ to any node $v \in \mathcal{I}$, and for border structure as the number of edges from $u$ to any node $v \in \mathcal{E}$.

### 4.3 Implementation and Model Extensions

Below we outline the computational complexity of our SUBGNN implementation and describe its extensions. Details on the training setup and hyperparameters, including a sensitivity analysis of new hyperparameters, are found in Appendix D and E.

**Computational complexity and runtime analysis.** The complexity of SUBGNN scales as the number and size of subgraphs grow. Notably, a fixed number of anchor patches is sampled, which upper bounds the memory and time complexity of the message passing algorithm. Structural anchor patch representations are generated via fixed-length random walks and are only calculated for a small number of anchor patches shared across all subgraphs. We precomputed similarities to speed up hyperparameter optimization, and find that pre-computation results in an 18X training speedup. With pre-computation, training SUB-GNN on the real-world dataset PPI-BP takes 30s per epoch on a single GTX 1080 GPU, of which 25%, 43%, and 32% are spent on neighborhood, position, and structure channels, respectively. Similarity calculation run time could be improved further through approximation methods such as locality sensitive hashing or $k$-hop shortest path distances.

**Model extensions.** Our implementation of SUBGNN initializes component embeddings with pre-trained node embeddings, which could alternatively be learned end-to-end. While we specify similarity functions for each channel, we note that $\gamma_x$ can be replaced with any function measuring the relevance of anchor patches to subgraph components, including learnable similarity functions. Finally, while this paper focuses on subgraph classification, the representation learning modules we present can be trivially extended to any unsupervised, semi-supervised, or other prediction tasks, simply by modifying the loss functions. SUBGNN can also be integrated into other frameworks as a sub-module for end-to-end training.

## 5 Experiments

We first describe our novel datasets, which consist of an underlying base graph and subgraphs with their associated labels. We then describe the baseline approaches and the experimental setup. Finally we present experiments on four synthetic and four real-world datasets for subgraph classification.

**Synthetic datasets.** We construct four datasets with subgraphs of specific graph properties such that binned values of the corresponding metrics act as their labels: DENSITY, CUT RATIO, CORENESS, and COMPONENT. DENSITY challenges the ability of our methods to capture the internal structure of our subgraphs; CUT RATIO, border structure; CORENESS, defined by the average core number of the subgraph, tests border structure and position; and COMPONENT, the number of subgraph components, tests internal and external position. The resulting DENSITY and CUT RATIO datasets have 250 subgraphs of size 20; CORENESS, 221 subgraphs of size 20; and COMPONENT, 250 subgraphs with 15 nodes per component. See Appendix B for more information on the methods used to add subgraphs and basic properties of the datasets.

**PPI-BP dataset.** PPI-BP is a real-world molecular biology dataset. Given a group of genes known to associate with some common biological process, we predict their collective cellular function. The base graph of the PPI-BP dataset is a human protein-protein interaction (PPI) network [85], and subgraphs are collections of proteins in the network involved in the same biological process (*e.g.*, "alcohol biosynthetic process," "mRNA cleavage," etc.). Protein subgraphs are labelled according to their cellular function from six categories (*e.g.* "metabolism", "development", etc.).

**HPO-METAB and HPO-NEURO datasets.** HPO-METAB and HPO-NEURO are real-world clinical diagnostic tasks for rare metabolic disorders and neurology, designed to mimic the task of rare disease diagnosis [56, 12, 4]. Given a collection of phenotypes intelligently sampled from a rare disease database, the task is to predict the subcategory of metabolic or neurological disease most

consistent with those phenotypes. The base graph is a knowledge graph containing phenotype and genotype information about rare diseases, and subgraphs consist of collections of phenotypes associated with a rare monogenic disease. HPO-METAB subgraphs are labelled according to the type of metabolic disorder [26, 34, 42], and HPO-NEURO subgraphs are labelled with one or more neurological disorders (multilabel classification) [26, 34].

**EM-USER dataset.** This is a user profiling task. Given the workout history of a user represented by that user's subgraph, we want to predict characteristics of that user. The base graph is a social fitness network from Endomondo [48], where the nodes represent workouts, and edges exist between workouts completed by multiple users. Each subgraph is represented as an Endomondo subnetwork that make up the user's workout history, and the labels are characteristics about the user (here, gender). See Appendix B for basic properties of all datasets.

**Alternative baseline approaches.** We consider seven baseline methods that can be applied to training subgraph embeddings. (1) AVG computes the average of the pre-trained node embeddings for nodes in each subgraph. (2) MN-GIN and (3) MN-GAT train meta nodes (also known as virtual nodes) [22, 28, 36] for each subgraph alongside the pre-training of the node embeddings [68, 62]. Both (4) S2V-N and (5) S2V-S compute subgraph embeddings using Sub2Vec [2], each separately preserving the neighborhood and structural properties, respectively. (6) S2V-NS concatenates the subgraph embeddings from S2V-N and S2V-S. Finally, (7) GC is a graph classification GNN (GIN) [19, 68] that aggregates node features from trainable node embeddings to produce subgraph embeddings; for GC, each subgraph is treated as a standalone graph.

**Implementation details.** For each dataset, we first train either GIN [68] or GraphSAINT [77] on link prediction to obtain node and meta node embeddings for the base graph. Sub2Vec subgraph embeddings are trained using the authors' implementation [2]. For a fair comparison, the feedforward component of all methods, including our own, was implemented as a 3-layer feed forward network with ReLu nonlinear activation and dropout. The feedforward networks were implemented with and without trainable node and subgraph embeddings, and the best results were reported for each. See Appendix D for more training details.

# 6 Results

Table 2: Micro-F1 on synthetic datasets. Standard deviations are provided from runs with 10 random seeds. Here SUBGNN is implemented with GIN node embeddings. See Appendix C for ROC scores. 'N' and 'S' stand for 'Neighborhood' and 'Structure,' respectively.

| Method | DENSITY | CUT RATIO | CORENESS | COMPONENT |
|---|---|---|---|---|
| SUBGNN (Ours) | **0.919**±**0.016** | **0.629**±**0.039** | **0.659**±**0.092** | **0.958**±**0.098** |
| Node Averaging | 0.429±0.041 | 0.358±0.055 | 0.530±0.050 | 0.516±<0.001 |
| Meta Node (GIN) | 0.442±0.052 | 0.423±0.057 | 0.611±0.050 | 0.784±0.046 |
| Meta Node (GAT) | 0.690±0.021 | 0.284±0.052 | 0.519±0.076 | 0.935±<0.001 |
| Sub2Vec Neighborhood | 0.345±0.066 | 0.339±0.058 | 0.381±0.047 | 0.568±0.039 |
| Sub2Vec Structure | 0.339±0.036 | 0.345±0.121 | 0.404±0.097 | 0.510±0.013 |
| Sub2Vec N & S Concat | 0.352±0.071 | 0.303±0.062 | 0.356±0.050 | 0.568±0.021 |
| Graph-level GNN | 0.803±0.039 | 0.329±0.073 | 0.370±0.091 | 0.500±0.068 |

**Synthetic datasets.** Results are shown in Tables 2 and 7. We find that our model, SUBGNN, significantly outperforms all baselines by 77.4% and the strongest baseline by 18.4%, on average. Results illustrate relative strengths and weaknesses of baseline models. While the GC method performs quite well on DENSITY (internal structure), as expected, it performs poorly on datasets requiring a notion of position or border connectivity. The MN baselines, in turn, performed best on COMPONENT, presumably because the meta node allows for explicit connection across subgraph components. Sub2Vec, the most directly related prior work, notably did not perform well on any task. This is unsurprising, as Sub2Vec does not explicitly encode position or border structure.

**Channel ablation.** To investigate the ability of our channels to encode their intended properties, we performed an ablation analysis over the channels (Table 4). Results show that performance of individual channels aligns closely with their inductive biases; for example, the structure channel

Table 3: Micro F1 on real-world datasets. Standard deviations are provided from runs with 10 random seeds. We report SUBGNN performance with both GIN and GraphSAINT node embeddings. See Appendix C for ROC scores. 'N' and 'S' stand for 'Neighborhood' and 'Structure,' respectively.

| Method | PPI-BP | HPO-NEURO | HPO-METAB | EM-USER |
|---|---|---|---|---|
| SUBGNN (+ GIN) | **0.599**$_{\pm 0.024}$ | 0.632$_{\pm 0.010}$ | **0.537**$_{\pm 0.023}$ | 0.814$_{\pm 0.046}$ |
| SUBGNN (+ GraphSAINT) | 0.583$_{\pm 0.017}$ | **0.644**$_{\pm 0.019}$ | 0.428$_{\pm 0.035}$ | 0.816$_{\pm 0.040}$ |
| Node Averaging | 0.297$_{\pm 0.027}$ | 0.490$_{\pm 0.059}$ | 0.443$_{\pm 0.063}$ | 0.808$_{\pm 0.138}$ |
| Meta Node (GIN) | 0.306$_{\pm 0.025}$ | 0.233$_{\pm 0.086}$ | 0.151$_{\pm 0.073}$ | 0.480$_{\pm 0.089}$ |
| Meta Node (GAT) | 0.307$_{\pm 0.021}$ | 0.259$_{\pm 0.063}$ | 0.138$_{\pm 0.034}$ | 0.471$_{\pm 0.048}$ |
| Sub2Vec Neighborhood | 0.306$_{\pm 0.009}$ | 0.211$_{\pm 0.068}$ | 0.132$_{\pm 0.047}$ | 0.520$_{\pm 0.090}$ |
| Sub2Vec Structure | 0.306$_{\pm 0.021}$ | 0.223$_{\pm 0.065}$ | 0.124$_{\pm 0.025}$ | **0.859**$_{\pm 0.014}$ |
| Sub2Vec N & S Concat | 0.309$_{\pm 0.023}$ | 0.206$_{\pm 0.073}$ | 0.114$_{\pm 0.021}$ | 0.522$_{\pm 0.043}$ |
| Graph-level GNN | 0.398$_{\pm 0.058}$ | 0.535$_{\pm 0.032}$ | 0.452$_{\pm 0.025}$ | 0.561$_{\pm 0.059}$ |

performs best on CUT RATIO (a border structure task) and DENSITY (an internal structure task), and the position channel performs best on COMPONENT (an internal position task). The CORENESS task primarily measures internal structure but also incorporates notions of border position; we find that the structure channel and combined channels perform best on this task.

Table 4: Channel ablation analyses (Micro F1). Channels that encode properties relevant to each dataset have a ✔ and best performing channels are in **bold**. We see in each case that the channels designed to encode relevant properties yield the best performance on each dataset. This suggests that the SUBGNN channels successfully encode their desired properties.

| SUBGNN Channel | DENSITY | CUT RATIO | CORENESS | COMPONENT |
|---|---|---|---|---|
| Position (P) | 0.758$_{\pm 0.046}$ | 0.516$_{\pm 0.083}$ | 0.581$_{\pm 0.044}$ ✔ | **0.958**$_{\pm 0.098}$ ✔ |
| Neighborhood (N) | 0.777$_{\pm 0.057}$ | 0.313$_{\pm 0.087}$ | 0.485$_{\pm 0.075}$ | 0.823$_{\pm 0.089}$ |
| Structure (S) | **0.919**$_{\pm 0.016}$ ✔ | **0.629**$_{\pm 0.039}$ ✔ | **0.663**$_{\pm 0.058}$ ✔ | 0.600$_{\pm 0.170}$ |
| All (P+N+S) | 0.894$_{\pm 0.025}$ | 0.458$_{\pm 0.101}$ | 0.659$_{\pm 0.092}$ | 0.726$_{\pm 0.120}$ |

**Real-world datasets.** Results are shown in Tables 3 and 8. We find that our model performs strongly on real-world datasets, outperforming baselines by 125.2% on average and performs 21.2% better than the strongest baseline. We experiment with both GIN and GraphSAINT node embeddings and find that each yields performance gains on different datasets. SUBGNN performed especially well relative to baselines on the HPO-NEURO and HPO-METAB tasks. These tasks are uniquely challenging because they require distinguishing subcategories of similar diseases (a challenge for averaging-based methods), exhibit class distributional shift between train and test, and have been designed to require inductive inference to nearby phenotypes using edges in the graph. Strong performance of SUBGNN on these tasks thus suggests the ability of our model to leverage its relational inductive biases for more robust generalization. Further results studying the generalizability of SUBGNN are in Appendix C. One baseline, S2V-S, achieved a higher micro F1 score, but lower AUC compared to SUBGNN on the EM-USER dataset. Finally, our model outperformed all baselines on the PPI-BP task, which like the HPO tasks operates over non-trivial biological subgraphs. Taken together, all of our new datasets are non-trivial and present unique challenges that highlight the relative strengths and weakness of existing methods in the field.

## 7 Conclusion

We present SUBGNN, a novel method for subgraph representation learning and classification, along with a framework outlining the key representational challenges facing the field. SUBGNN performs subgraph level message passing with property-aware channels. We present eight new datasets (4 synthetic, 4 real-world) representing a diverse collection of tasks that we hope will stimulate new subgraph representation learning research. SUBGNN outperforms baselines by an average of 77.4% on synthetic datasets and 125.2% on real-world datasets.

## Broader Impact

**A variety of impactful application areas.** The ability of SUBGNN to learn powerful subgraph representations creates fundamentally new opportunities for applications beyond the reach of node-, edge-, and graph-level tasks. Such applications require us to be able to reason about subgraphs and predict properties of subgraphs by leveraging the fact that subgraphs reside within a large, underlying graph. For example, this work was directly motivated by the challenge of rare disease diagnosis, which motivated the generation of two new datasets that we are releasing specifically to foster methods that will eventually prove useful on subgraph predict tasks (*e.g.*, patient subgraphs residing within a large biomedical knowledge graph). Likewise, drug development represents another broad potential area of research that is closely connected with the field of graph neural networks. Hence, we also developed a new bioinformatics dataset. In addition, there are many other potentially positive applications for this class of method, including the prediction of toxic behavior on social media.

**The need for thoughtful use of SUBGNN framework.** We also recognize that Subgraph Neural Networks have potential to used for harmful applications. For example, the potential benefit of predicting toxic communities on social media is inextricably tied with the capacity to leverage these same networks for malicious political and social purposes. Another broad class of harms that could involve subgraph classification include those harms that emerge from high-resolution user profiling. As with any data-driven methods, there is also an opportunity for bias to exist at all stages of the model development process. In the case of biomedical data science, for example, biases can exist within the data itself, and disparities can exist both in their efficacy as well as their deployment reach.

**Real-world subgraph datasets.** The release of our monogenic disease and bioinformatics datasets is, in large part, motivated by our desire to help steer the community towards beneficial rather than malicious applications of these tools. Ultimately, given the relative immaturity of this field among major areas of computer science, vigilance is required on the part of researchers (and other relevant experts) to ensure that we as a community pursue the best possible use of our tools.

## Acknowledgments and Disclosure of Funding

S.G.F. was supported by training grant T32GM007753 from the National Institute of General Medical Science, NIH. M.M.L. was supported by T32HG002295 from the National Human Genome Research Institute, NIH. M.Z. is supported, in part, by NSF grant nos. IIS-2030459 and IIS-2033384, and by the Harvard Data Science Initiative. The content is solely the responsibility of the authors.

## Footnotes

[2]Code and datasets are available at https://github.com/mims-harvard/SubGNN.

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
