[Supplementary Material]

# Appendix A  Further Details on SUBGNN

In this section, we outline additional details and design decisions for the implementation of SUBGNN. In Section A.1, we provide an algorithmic overview of our method, and in Section A.2, we provide further motivation and details on triangular random walks and the structural anchor patch embedder algorithm.

## A.1  Message Passing Algorithm

For brevity, Algorithm 1 summarizes the forward pass of SUBGNN for a single subgraph $S$. While the algorithm demonstrates how to embed subgraph $S$, we want to emphasize that SUBGNN learns representations for every subgraph $S \in \{S_1, S_2, \ldots, S_n\}$. In practice, we extend Algorithm 1 to multiple subgraphs using mini-batching. Note that any aggregation function that operates over an unordered set of vectors can be used for $\text{AGG}_M$, $\text{AGG}_C$, $\text{AGG}_L$, and/or READOUT. Our implementation leverages the sum operator for $\text{AGG}_M$ and READOUT and the concat operator for $\text{AGG}_C$ and $\text{AGG}_L$. We empirically found that applying attention [72] over the individual subgraph component representations did not yield any performance gains over summation for READOUT.

---

**Algorithm 1:** SUBGRAPH NEURAL NETWORK. Channels N, S, and P correspond to neighborhood, structure, and position. Subchannels I and B correspond to internal and border subgraph topology.

---

**Input:** Graph $G = (V, E)$; Node representations $\{\mathbf{x}_u | u \in V\}$; Subgraph $S$ consisting of
  connected components $S^{(c)}$ for $c = 1, \ldots, n_c$; Anchor patch sampler $\phi_X$, anchor patch encoder
  $\psi_X$, and anchor patch similarity function $\gamma_X$ for each channel X; Nonlinear activation function $\sigma$
**Output:** Subgraph representation $\mathbf{h}_S$ for subgraph $S$
**Model Parameters:** Layer-wise, channel-specific learnable weight matrices $\mathbf{W}_X$ and $\mathbf{Q}_X$

$\mathbf{z}_c^{(0)} = \sum_{u \in S^{(c)}} \mathbf{x}_u$
$\mathbf{h}_{X,c}^{(0)} = \mathbf{z}_c^{(0)}$ for channel $X \in \{N, S, P\}$      // Channel-independent initialization
**for** *layer $l = 1, \ldots, L$* **do**
   $A_X^{(i)} = \phi_X(G)$ for $i = 1, \ldots, n_A$ and channel $X \in \{S_I, S_B, P_B\}$     // See Section 4.2
   $A_X^{(i)} = \phi_X(G, S)$ for $i = 1, \ldots, n_A$ and channel $X \in \{P_I\}$
   **for** *connected component $c = 1, \ldots, n_c$* **do**
     $A_X^{(i)} = \phi_X(G, S^{(c)})$ for $i = 1, \ldots, n_A$ and channel $X \in \{N_I, N_B\}$
     **for** *channel $X \in \{\{P_I, P_B\}, \{N_I, N_B\}, \{S_I, S_B\}\}$* **do**
       **for** *anchor patch $i = 1, \ldots, n_A$* **do**
         $\mathbf{a}_X^{(i)} = \psi_X(A_X^{(i)})$                    // E.g., Algorithm 2
         $\mathbf{m}_{X,c,i}^{(l)} = \text{MSG}_X^{A_X^{(i)} \to S^{(c)}}$             // (Eq. 1)
         $\mathbf{M}_{X,c}^{(l)}[i] = \mathbf{m}_{X,c,i}^{(l)}$ if $X \in \{S_*, P_*\}$
       **end**
       $\mathbf{z}_{X,c}^{(l)} = \sigma(\mathbf{Q}_X^{(l)} \cdot \mathbf{M}_{X,c}^{(l)})$ if $X \in \{S_*, P_*\}$     // Property-aware output rep.
       $\mathbf{g}_{X,c}^{(l)} = \text{AGG}_M(\{\mathbf{m}_{X,c,1}^{(l)}, \ldots, \mathbf{m}_{X,c,n_A}^{(l)}\})$    // Aggregate messages (Eq. 2)
       $\mathbf{h}_{X,c}^{(l)} = \sigma(\mathbf{W}_X^{(l)} \cdot [\mathbf{g}_{X,c}^{(l)}; \mathbf{h}_{X,c}^{(l-1)}])$     // Order-invariant rep. (Eq. 2)
     **end**
     $\mathbf{z}_c^{(l)} = \text{AGG}_C(\mathbf{h}_{N,c}^{(l)}, \mathbf{z}_{S,c}^{(l)}, \mathbf{z}_{P,c}^{(l)})$            // Aggregate channels
   **end**
**end**
$\mathbf{z}_c \leftarrow \text{AGG}_L(\{\mathbf{z}_c^{(0)}, \ldots, \mathbf{z}_c^{(L)}\})$             // Aggregate layers
$\mathbf{h}_S = \text{READOUT}(\{\mathbf{z}_1, \ldots, \mathbf{z}_{n_c}\})$        // Aggregate subgraph components

---

### A.2 Triangular Random Walks

In Section 4.2, we leverage triangular random walks [9] to (1) sample structure anchor patches via $\phi_\text{s}$ and (2) embed the structure anchor patches via $\psi_\text{s}$. Refer to Algorithm 2 for a summary of the structure anchor patch embedder.

Triangular random walks are parameterized by $\beta \in [0, 1]$ and yield a sequence of nodes $X_0, X_1, \ldots, X_n$. The parameter $\beta$ determines whether triangles or non-triangles will be privileged during sampling. A node $z$ is said to be a triangular successor to nodes $x$ and $y$ if $x, y, z$ would form a triangle. The triangular random walk samples triangular successors with probability $\beta$ and non-triangular successors with probability $1 - \beta$. When performing an internal random walk, we initialize $P[X_0 = x] = 1/|\mathcal{I}|$ and $P[X_1 = y | X_0 = x] = 1/d_I(x)$, where $d_I(x)$ is number of edges from $x$ to other nodes in $\mathcal{I}$. Similarly, we initialize a border random walk with $P[X_0 = x] = 1/|\mathcal{B}|$ and $P[X_1 = y | X_0 = x] = 1/d_E(X_0)$ where $d_E(x)$ is number of edges from $x$ to nodes in $\mathcal{E}$.

We then define the transition probability for any subsequent step of the random walk as follows: given current node $y$ and preceding node $x$, the probability of transition to node $z$ is defined as $P[X_{t+1} = z | X_t = y, X_{t-1} = x] = [\beta(1/|T|)\mathbb{1}_{X_{t+1} \in T} + (1 - \beta)(1/|U|)\mathbb{1}_{X_{t+1} \in U}$ where $T$ is the set of triangular successors of nodes $x$ and $y$ and $U$ is the set of non-triangular successors. More precisely, $T(X_t, X_{t-1}) = N(X_t) \cap N(X_{t-1})$ and $U(X_t, X_{t-1}) = N(X_t) \setminus N(X_{t-1})$.

---

**Algorithm 2:** STRUCTURE ANCHOR PATCH EMBEDDER.

---

**Input:** Graph $G$; Anchor patch $A$; Node representations $\{\mathbf{x}_u | u \in A\}$; Triangle random walk (TRIANGLE_RW) parameter $\beta$; Number of walks $n$; Length of walks $k$
**Output:** Representation $\mathbf{a}$ for anchor patch $A$
**for** $i = 1, \ldots, n$ **do**
$\quad | \quad (u_{\pi(1)}, \ldots, u_{\pi(k)}) = \text{TRIANGLE\_RW}(G, A, \beta)$
$\quad | \quad \mathbf{h}_i = \text{BI-LSTM}((\mathbf{x}_{\pi(1)}, \ldots, \mathbf{x}_{\pi(k)}))$ where $\mathbf{x}_{\pi(j)}$ is representation for node $u_{\pi(j)}$
**end**
$\mathbf{a} = \sum_{i=1,\ldots,n} \mathbf{h}_i$

---

## Appendix B   Further Details on Datasets

We proceed by describing the construction and processing of synthetic as well as real-world datasets. Note that we provide all datasets in our SUBGNN code release.

### B.1   Synthetic Datasets for Subgraph Classification

**Generating base graphs.** For DENSITY, CUT RATIO, and COMPONENT, we start with a base Barabási-Albert graph, where the number of preferentially attached edges, $m$, is 5 for DENSITY and CUT RATIO and $m = 1$ for DIAMETER and COMPONENT. In CORENESS, the base graph is a duplication-divergence graph where the probability of retaining the edge of the replicated node is 0.7. Refer to Table 5 for properties of the base graphs.

**Generating subgraphs.** We introduce three methods for generating subgraphs: (1) the PLANT method, which searches for $n > 1$ common nodes and $e \geq 1$ shared edges between the base graph and the new subgraph, and plants the subgraph on the base graph via the union of the node and edge sets; (2) the STAPLE method, which adds an edge between a randomly sampled node in the base graph and a node in the new subgraph; and (3) the BFS method, which creates subgraphs in the base graph by performing breadth first search with a specified max depth $d$ from randomly selected start nodes. Figure 3 describes the PLANT and STAPLE approaches in more detail.

We construct subgraphs with various desired graph properties: density, cut ratio, $k$-coreness, and number of components [71, 55]. Density is defined as $D = (2 \cdot |E|)/(|V| \cdot |V - 1|)$ for our undirected graphs. We define cut ratio as the proportion of edges shared between the subgraph $S$ and the rest of the graph $G$, specifically $\text{CR}(S) = |\mathcal{B}_S|/(|S||G \setminus S|)$, where $\mathcal{B}_S = \{(u, v) \in E | u \in S, v \in G \setminus S\}$. The $k$-core of a graph is the maximum subgraph with a minimum degree of at least $k$. A connected component is defined as a set of nodes in a subgraph such that each pair of nodes is connected by a path.

| **(a) Base graph and new subgraph** | **(b) Planting** | **(c) Stapling** |

● ● ● ●   **Nodes** in graph (colors corresponding to a different subgraph)
———   **Edges** in graph
○   **Shared nodes** between base graph and new subgraph
———   **Shared edges** between base graph and new subgraph

Figure 3: Procedures for generating synthetic graphs. **(a)** Base graph and new subgraph to be added. **(b)** New synthetic graph with a "planted" subgraph. **(c)** New synthetic graph with a "stapled" subgraph. The base graph **(a)** has 3 subgraphs, and the new graphs **(b)-(c)** each have 4 subgraphs, indicated by the colors red, light blue, light purple, and dark purple. The yellow nodes and bright red edges are those shared between the base graph and the new subgraph.

**Subgraph classification tasks.** For each subgraph classification task, we generate subgraphs that vary along a specific network property (e.g. density, cut ratio). Labels are then generated by binning the network property values into two or three quantiles: (1) DENSITY has 250 subgraphs that are constructed via BFS with $d = 3$ to have low, medium, or high density. (2) CUT RATIO has 250 subgraphs constructed by PLANTING complete graphs onto the base graph such that the subgraphs have low, medium, or high cut ratio. (3) CORENESS has 221 subgraphs that are created using the duplication-divergence model and constructed via PLANTING onto the base graph. These subgraphs' labels (*e.g.*, low, medium, or high coreness bins) are assigned by calculating the average $k$-coreness of all nodes in the subgraph. Finally, (4) COMPONENT contains 250 Barabási-Albert subgraphs that are STAPLED onto the base graph, and the associated label is the number of connected components in the subgraph (single or multiple components). Subgraphs for synthetic datasets are split into train, validation, and test sets via a 50/25/25 split. See Table 6 for properties of subgraphs in the datasets.

## B.2 Novel, Real-World Datasets for Subgraph Classification

**PPI-BP dataset.** The base graph of the PPI-BP dataset is a human protein-protein interaction (PPI) network [85]. Nodes represent human proteins specified by their Entrez IDs [38], and edges exist between nodes if there is physical interaction between the proteins. Subgraphs are collections of proteins in the PPI network involved in the same biological process (*e.g.*, "alcohol biosynthetic process," "mRNA cleavage," etc.). These subgraphs were obtained from the Gene Ontology (GO) gene sets from the Molecular Signatures Database (MSigDB) [57]. PPI-BP subgraphs labels are obtained from the "GO Slim" resource in the GO Biological Process Ontology [14, 3], which groups narrow processes into broader categories: metabolism, development, signal transduction, stress/death, cell organization, and transport. Gene sets are limited to those containing at least five genes, and we exclude any gene sets that comprised 70% or more of the genes in any other gene set. Subgraphs are split 80/10/10 at random into training, validation, and test sets. There are on average 10.2 nodes per subgraph and 265.2 subgraphs across the 6 labels [57].

**HPO-NEURO dataset.** The base graph is a knowledge graph containing phenotype and genotype information about rare diseases. Nodes are phenotypes (symptoms), and edges exist between phenotypes if (1) they are caused by a mutation in a shared gene according to DisGeNET [50], HPO-A [34], or Orphanet [39] or (2) an edge exists between the phenotypes according to the Human Phenotype Ontology (HPO) [52]. Each subgraph consists of a set of phenotypes associated with a rare monogenic disease. The subgraphs contain noisy phenotypes unrelated to the disease, distractor phenotypes related to incorrect but similar diseases, and less specific phenotypes generated by walking up the HPO hierarchy (*e.g.*, from "arachnodactyly" to "abnormality of the fingers"). Together, these simulate the imperfect diagnosis process and make the diagnosis task more realistic. Subgraph labels are the diagnosis categories. This is a multi-label dataset consisting of 10 neurological

Table 5: Properties of **base graphs** in synthetic and real-world datasets.

| Dataset | # Nodes | # Edges | Density | # Subgraphs | # Labels |
|---|---|---|---|---|---|
| DENSITY | 5,000 | 29,521 | 0.0024 | 250 | 3 |
| CUT RATIO | 5,000 | 83,969 | 0.0067 | 250 | 3 |
| CORENESS | 5,000 | 118,785 | 0.0095 | 221 | 3 |
| COMPONENT | 19,555 | 43,701 | 0.0002 | 250 | 2 |
| PPI-BP | 21,521 | 342,316 | 0.0015 | 1,591 | 6 |
| HPO-METAB | 14,587 | 3,238,174 | 0.0304 | 2,400 | 6 |
| HPO-NEURO | 14,587 | 3,238,174 | 0.0304 | 4,000 | 10 |
| EM-USER | 57,333 | 4,573,417 | 0.0028 | 324 | 2 |

Table 6: Properties of **subgraphs** in synthetic and real-world datasets.

| Dataset | Average # nodes | Average density | Average cut ratio | Average # components |
|---|---|---|---|---|
| DENSITY | $20.0_{\pm 0.0}$ | $0.232_{\pm 0.146}$ | $0.0010_{\pm 0.0062}$ | $3.8_{\pm 3.7}$ |
| CUT RATIO | $20.0_{\pm 0.0}$ | $0.945_{\pm 0.028}$ | $0.0072_{\pm 0.0011}$ | $1.0_{\pm 0.0}$ |
| CORENESS | $20.0_{\pm 0.0}$ | $0.219_{\pm 0.062}$ | $0.0082_{\pm 0.0081}$ | $1.0_{\pm 0.0}$ |
| COMPONENT | $74.2_{\pm 52.8}$ | $0.150_{\pm 0.161}$ | $5.1 \times 10^{-6}{}_{\pm 3.4 \times 10^{-6}}$ | $4.9_{\pm 3.5}$ |
| PPI-BP | $10.2_{\pm 10.5}$ | $0.216_{\pm 0.188}$ | $0.0030_{\pm 0.0026}$ | $7.0_{\pm 5.5}$ |
| HPO-METAB | $14.4_{\pm 6.2}$ | $0.757_{\pm 0.149}$ | $0.1844_{\pm 0.0396}$ | $1.6_{\pm 0.7}$ |
| HPO-NEURO | $14.8_{\pm 6.5}$ | $0.767_{\pm 0.141}$ | $0.1834_{\pm 0.0386}$ | $1.5_{\pm 0.7}$ |
| EM-USER | $155.4_{\pm 100.2}$ | $0.010_{\pm 0.006}$ | $0.0053_{\pm 0.0006}$ | $52.1_{\pm 15.3}$ |

disease categories: neurodegenerative, epilepsy, ataxia, genetic dementia, central nervous system malformation, intellectual, neurometabolic, movement, peripheral neuropathy, and neuromuscular disease. Subgraphs are split by disease into train, validation, and test sets via an 80/10/10 split.

**HPO-METAB dataset.** This is a clinical diagnostic task for rare metabolic disorders, defined similarly to HPO-NEURO, but for a different collection of diseases and disease categories. The base graph is identical to the base graph in HPO-NEURO, and the subgraphs consist of a set of phenotypes associated with a rare monogenic metabolic disease. The HPO-METAB dataset contains 6 labels corresponding to types of metabolic disease: lysosomal, energy, amino acid, carbohydrate, lipid, and glycosylation. Subgraphs are split by disease into train, validation, and test sets via an 80/10/10 split.

**EM-USER dataset.** The Endomondo base graph is a co-occurrence network: nodes represent workouts, and edges exist between workouts completed by multiple users. As such, the graph contains cliques (*i.e.*, small fully connected networks) of highly popular combinations of workouts. We identify co-occurrence cliques and use random network sampling to break up cliques in the base graph [32, 51]. Examples are split 70/15/15 by workout into train, validation, and test sets.

See Table 5 and 6 for further properties of all datasets.

## Appendix C  Details on Empirical Evaluation

Baseline models and SUBGNN were evaluated using Micro F1 and AUROC. Both metrics were implemented using Scikit-learn (Version 0.20.2). AUROC scores for synthetic datasets are in Table 7, and results for real-world datasets are in Table 8. AUROC scores for the SUBGNN channel ablation analysis are in Table 9.

**Generalizability analysis.** Generalizability in subgraph representation learning is an interesting area of future research, in part because in this context it could be defined in several competing ways. For example, one aspect of generalizability in subgraph prediction is the ability to make predictions about subgraphs that contain *nodes* that were never seen during training. To probe this aspect of SUBGNN, we measure the test performance of the model as a function of the overlap in nodes between train and test subgraphs, irrespective of whether the nodes were participating in similar structures or whether the labels between the subgraphs were also shared.

Table 7: AUROC performance on synthetic datasets. Standard deviations are provided from runs with 10 random seeds. 'N' and 'S' stand for 'Neighborhood' and 'Structure,' respectively.

| | Datasets | | | |
|---|---|---|---|---|
| Method | DENSITY | CUT RATIO | CORENESS | COMPONENT |
| SUBGNN (Ours) | **0.971**±**0.007** | **0.836**±**0.021** | **0.824**±**0.044** | **0.997**±**0.009** |
| Node Averaging | 0.619±0.026 | 0.542±0.069 | 0.700±0.028 | 0.623±0.199 |
| Meta Node (GIN) | 0.602±0.039 | 0.602±0.023 | 0.670±0.047 | 0.873±0.026 |
| Meta Node (GAT) | 0.809±0.024 | 0.531±0.089 | 0.682±0.068 | 0.884±0.014 |
| Sub2Vec Neighborhood | 0.580±0.028 | 0.533±0.046 | 0.592±0.037 | 0.629±0.035 |
| Sub2Vec Structure | 0.553±0.026 | 0.504±0.093 | 0.539±0.084 | 0.539±0.104 |
| Sub2Vec N & S Concat | 0.578±0.040 | 0.493±0.051 | 0.562±0.043 | 0.558±0.042 |
| Graph-level GNN | 0.868±0.069 | 0.494±0.045 | 0.697±0.113 | 0.690±0.308 |

Table 8: AUROC performance on real-world datasets. Standard deviations are provided from runs with 10 random seeds. 'N' and 'S' stand for 'Neighborhood' and 'Structure,' respectively.

| | Datasets | | | |
|---|---|---|---|---|
| Method | PPI-BP | HPO-NEURO | HPO-METAB | EM-USER |
| SUBGNN (+ GIN) | **0.816**±**0.012** | 0.862±0.005 | **0.843**±**0.014** | 0.911±0.042 |
| SUBGNN (+ GraphSAINT) | 0.797±0.008 | **0.863**±**0.011** | 0.771±0.027 | **0.947**±**0.009** |
| Node Averaging | 0.498±0.009 | 0.764±0.104 | 0.814±0.032 | 0.896±0.143 |
| Meta Node (GIN) | 0.474±0.006 | 0.516±0.044 | 0.510±0.036 | 0.536±0.082 |
| Meta Node (GAT) | 0.535±0.017 | 0.502±0.012 | 0.581±0.017 | 0.485±0.056 |
| Sub2Vec Neighborhood | 0.518±0.013 | 0.502±0.014 | 0.504±0.039 | 0.496±0.108 |
| Sub2Vec Structure | 0.551±0.016 | 0.498±0.010 | 0.505±0.016 | 0.936±0.008 |
| Sub2Vec N & S Concat | 0.544±0.011 | 0.504±0.010 | 0.496±0.015 | 0.518±0.048 |
| Graph-level GNN | 0.663±0.044 | 0.773±0.027 | 0.772±0.018 | 0.525±0.065 |

Notably, test subgraphs in the COMPONENT and CORENESS datasets have zero nodes in common with any train or validation subgraphs, yet SUBGNN performs strongly on both datasets (Table 2). Figure 4 shows Micro F1 performance as a function of node overlap on one randomly selected real-world dataset, HPO-METAB. While SUBGNN performs considerably better than majority class and random baselines on the test subgraphs with the smallest percent overlap, it is clear that future research is needed to improve generalization performance.

## Appendix D   Implementation Details

**Computing infrastructure.** We leverage Pytorch Geometric (Version 1.4.3) [19] and Pytorch Lightning (Version 0.7.1) [17] for model development. Models were trained on single GPUs from a SLURM cluster containing Tesla V100, Tesla M40, Tesla K80, and GeForce GTX 1080 GPUs.

**Pretraining node embeddings.** Node embeddings were pretrained using a 2-layer GIN architecture [68]. Hyperparameters were selected from the following ranges: batch size $\in [256, 4096]$, learning

Table 9: Channel ablation analyses (AUROC). Channels that encode properties relevant to each dataset have a ✔ and best performing channels are in **bold**.

| | Datasets | | | |
|---|---|---|---|---|
| SUBGNN Channel | DENSITY | CUT RATIO | CORENESS | COMPONENT |
| Position (P) | 0.899±0.016 | 0.706±0.043 | 0.712±0.047 ✔ | **0.997**±**0.009** ✔ |
| Neighborhood (N) | 0.904±0.020 | 0.528±0.078 | 0.668±0.066 | 0.955±0.035 |
| Structure (S) | **0.971**±**0.007** ✔ | **0.836**±**0.021** ✔ | 0.823±0.025 ✔ | 0.834±0.142 |
| All (P+N+S) | 0.968±0.008 | 0.642±0.100 | **0.824**±**0.044** | 0.968±0.032 |

Figure 4: Micro F1 score as a function of maximum percent node overlap with any subgraph in the HPO-METAB train set. Majority class performance = 0.026, and random performance = 0.166. Bars represent standard deviation from runs with 10 random seeds.

rate $\in [5e\text{-}3, 5e\text{-}5]$, weight decay $\in [5e\text{-}4, 5e\text{-}5]$, dropout rate $\in [0.4, 0.5]$, hidden layer dimension $\in [128, 512]$, and output dimension $\in [32, 128]$. For all baselines and SUBGNN, we used NEIGH-BORSAMPLER [25] in Pytorch Geometric [19] to perform mini-batching with number of hops $k = 1$ and neighborhood size $\in [0.1, 1.0]$. To demonstrate that SUBGNN is not dependent on GIN and NEIGHBORSAMPLER, we replaced the GIN layers with GCN and implemented GRAPHSAINT for mini-batching given walk length $\in [16, 32]$ and number of steps $\in [16, 32]$ [33, 77]. The node features for all graphs were one-hot encodings.

**Model hyperparameter tuning.** Hyperparameters were selected to optimize micro F1 scores on the validation datasets. In the following paragraph, we describe the hyperparameter ranges we explored. The best hyperparameters selected for each model can be found at https://github.com/mims-harvard/SubGNN.

Baseline hyperparameters were selected from the following ranges: batch size $\in [8, 128]$, learning rate $\in [1e\text{-}5, 0.1]$, weight decay $\in [5e\text{-}5, 5e\text{-}6]$, and feed forward hidden dimension sizes $\in [8, 256]$. For the MN-GAT baseline method, we use the default parameters for GAT, except for the number of heads, which we set to 4. For the S2V-N and S2V-S, methods, we used all of the default parameters in Sub2Vec [2] to train subgraph embeddings; S2V-NS concatenates the resulting embeddings from S2V-N and S2V-S. The GC architecture and hyperparameters were adapted from the Pytorch Geometric GIN example on the MUTAG dataset (https://github.com/rusty1s/pytorch_geometric/blob/master/examples/mutag_gin.py) [19].

SUBGNN hyperparameters were selected from the following ranges: batch size $\in [16, 128]$, learning rate $\in [1e\text{-}4, 1e\text{-}3]$, gradient clipping $\in [0, 0.5]$, number of layers $l \in [1, 4]$, k-hop neighborhood $\in [1, 2]$, number of internal position anchor patches $|A_{\text{P}_I}| \in [25, 75]$, $|A_{\text{P}_B}| \in [50, 200]$, $|A_{\text{N}_I}| \in [10, 25]$, $|A_{\text{N}_B}| \in [25, 75]$, $|A_{\text{S}}| \in [15, 45]$, number of LSTM layers $\in [1, 2]$ with dropout $\in [0.0, 0.4]$, and feed forward hidden dimension sizes $\in [32, 64]$ with dropout $\in [0.0, 0.4]$. We additionally experimented with both sum and max aggregation for the connected component initialization from node embeddings, and we tested the Pytorch Lightning auto learning rate finder.

## Appendix E   Hyperparameter sensitivity analysis

We performed a hyperparameter sensitivity analysis to measure the dependence of SUBGNN on training hyperparameter configuration. Starting with the best performing model for the relevant channel, we vary one hyperparameter at a time and report validation performance on the HPO-

Figure 5: Sensitivity analysis of hyperparameters in SUBGNN. Varying most of the hyperparameters leads to < 0.05 change in Micro F1 score.

METAB dataset. To study model behavior at the extremes, we tested wider hyperparameter ranges: batch size $\in [16, 128]$, learning rate $\in [1e\text{-}6, 5e\text{-}2]$, gradient clipping $\in [0, 1]$, number of layers $l \in [1, 4]$, k-hop neighborhood $\in [1, 4]$, number of internal position anchor patches $|A_{\mathrm{P}_I}| \in [1, 100]$, $|A_{\mathrm{P}_B}| \in [10, 300]$, $|A_{\mathrm{N}_I}| \in [1, 75]$, $|A_{\mathrm{N}_B}| \in [1, 150]$, $|A_{\mathrm{S}}| \in [1, 120]$, number of LSTM layers $\in [1, 4]$ with dropout $\in [0.0, 0.8]$, and feed forward hidden dimension sizes $\in [8, 256]$. Of all hyperparameters in our model, we find that seven strongly impact performance (change in micro F1 >.05), two of which are common across all neural networks: learning rate, feed forward hidden dimension, connected component aggregation (sum or max), number of structure anchors, number of internal and border neighborhood anchors, and number of border position anchors.