[Reviews · NeurIPS 2020]

Review 1

Summary and Contributions: ===================== POST-REBUTTAL RESPONSE =================== The authors promised to address most of the concerns in my review. I am mostly concerned about their statement that SUB-GNN "proposes a new field of study: subgraph classification" since Meng et al, 2018 had already studied the same task. I will agree that this "is the first method that works with varying subgraph sizes", and "can easily deal with large subgraphs". You don't need to claim to be the first on everything, it weakens the other claims in your paper. It is OK to have an engineering advance on an important topic. A practical approach being grounded in network science theory is not equal to making a theoretical contribution on our understanding of subgraph classification using neural networks. The Monte Carlo procedure uses sampled lighthouses but it is not averaging these samples. Odd. The authors would likely see performance improvements with an averaging. All said, I think the paper should be accepted as long as the authors make the changes they have promised and add some discussion about the point I raised. ================================================ This work considers a graph representation learning method that, unlike GNNs, can give embeddings to subgraphs rather than just nodes. While not the first work on the topic, it is the first to give a method that work with varying subgraph sizes. It can also easily deal with large subgraphs. The method is a great engineering effort, drawing ideas from multiple sources. It is a bit of a kitchen sink approach, with little theory guiding the decisions.

Strengths: Theoretical grounding: This is an engineering effort that draws inspiration from a multitude of methods. Some of the decisions are arbitrary, but it does not detract from the engineering feat. Significance and novelty of the contribution: First method that can give subgraph representations to large variable-size subgraphs. It is not the first method designed to give representations to subgraphs (see Meng et al., 2018). Empirical evaluation: The work shows a lot of interesting tasks. Nice job.

Weaknesses: - It is not clear how the method can be inductive (applied to new graphs). It relies heavily on the subgraph anchor points. Are the embeddings averaged over all possible sampled anchor points? It is also unclear how the subgraph anchor points can be chosen in graphs of different sizes. In some of the smaller graphs, how do we ensure the anchor points do not overlap? And if they overlap, how well is the method supposed to work? - The work offers very little in terms of theory. A lot of decisions are made but it is hard to grasp why all these decisions came together into a single method. - What is about random walks that is so fundamental for the subgraph representation learning? - What is the effect of having node representations trained with a GNN *(GIN)? - Why not use a direct measure of graph similarity (e.g., edit distance) rather than triangle-biased *random walks* and degree sequences? And why Dynamic Time Warping as the base metric of these random-walk-chosen degree sequences? - What is the effect of the anchor point sampling procedure? - The method has a very large number of hyperparameters (the many subgraph choosing methods, the random walk parameters, the similarity measure, etc.). It is hard to think that one can devise a methodology to automatically choose most of these hyperparameter without facing serious problems with family-wise error rates. - The description of the experiments offers little assurance that there is no leakage of the test data into the training data, since the train and test subgraphs can overlap. "[The] subgraphs are divided into 80/10/10 split." is not reassuring to someone who is worried about leakage.

Correctness: - The paper claims that permutation invariance of h_{X,c} removes position-awareness. It can't really remove all position information. This statement needs to be more precise.

Clarity: The description of the method is hard to follow. In part because of typos and in part due to the notation, but.... - Most importantly, the only complete description of the method appears in the Appendix, in form of Algorithm 1. I was not able to understand the approach just reading the main paper (not even roughly). After spending a lot of time in the Appendix, I was able to grasp how it works. Main paper typos: The notation has bugs and is hard to follow: - S^{(C)} should be S^{(c)}? If MSG_X is not a function of C and A_x^{(q)}, how can we build H_{X,c} from it? - \cdot is used for matrix multiplication but maybe also Hadamard product? - \{{\bf z}_c \forall c \inS\} => is missing a comma: \{{\bf z}_c, \forall c \inS\} Appendix typos: Algorithm 1: S^{(C)} for c=1,...,R should be S^{(c)} ?? Algorithm 2: For m = 1, ..., M should be For w = 1,...,M ??

Relation to Prior Work: There is some prior work whose citation is missing. Specifically, this is not the first graph representation learning work to consider subgraph tasks. The earliest work I am aware of was (Meng et al., 2018), which I think only dealt with (small) fixed-size subgraphs. The approach was interpretable, which I thought was interesting at the time. Meng, C., Mouli, S.C., Ribeiro, B. and Neville, J., 2018, April. Subgraph pattern neural networks for high-order graph evolution prediction. In Thirty-Second AAAI Conference on Artificial Intelligence.

Reproducibility: No

Additional Feedback: I don't think there is much to fix with respect to my other comments, except: 1. Convince the reader that there is no leakage between test and train/validation. 2. How should all these hyperparameter be chosen without high family-wise error rates?


Review 2

Summary and Contributions: The paper proposes to consider subgraph classification as a new task for graph neural networks and proposes a model tailored to this task, as well as corresponding datasets. In a nutshell, the model is GNN that uses standard message passing in three channels (also, factors). Messages are passed within each subgraph and between a subgraph and sampled anchor nodes from the graph. The idea is interesting and definitely relevant to NeurIPS. Overall, the paper is well written, but it is very dense -- since the model proposed by the authors is rather involved and all the general explanations of the new tasks and datasets take space. Nevertheless, I think, the authors have done a great job and the paper provides useful contributions for the NeurIPS community. Overall, I think the submissions provides enough contribution to be accepted. Details are listed below. ================ I acknowledge the rebuttal. Since the other reviewers seem to have similar problems with readability and understanding, and pointed out several further errors, I decreased my scoring.

Strengths: - propose a new field of study: subgraph classification - the model itself is interesting - propose corresponding baselines and created datasets

Weaknesses: In my opinion, there are just smaller issues regarding readability, and minor clarifications are needed (mentioned below).

Correctness: To me it seems so.

Clarity: Generally yes. There is room of improvement regarding readability, I mention some points in the feedback section.

Relation to Prior Work: Yes.

Reproducibility: Yes

Additional Feedback: - Related Work: Is covered broadly. Since the method of the authors heavily relies on the three factors they propose, it would make sense to mention alternative factor representations of nodes/graphs in the literature. - Approach: The overall approach is interesting and makes sense to me. Yet, this section is very dense and many parts are confusing at first read. Maybe the authors could try to ease reading visually by using an enumeration in each paragraph in Sec. 4.2, with one item per channel. - Experiments: * The considered baselines are reasonable and cover a wide range of alternative methods. * Notably, the authors propose a set of datasets for subgraph classification. * The HPO tasks look slightly artificial to me. Although this is ok since you can use them for benchmarking, I would be interested if the authors know about if this kind of classification is actually done in practice, or if there's a need for such approaches. * The results show that the proposed model clearly outperforms existing methods and thus presents a good start in exploring subgraph classification as a new area. ------------------------------------------ Some minor issues: * l.180 add a \mid or comma after z_c * Table 1 is not self-explaining, e.g., "Identity of S’s internal nodes" and "Identity of S’s border nodes" are not directly clear. * l.258 "subgraphs are defined by the Biological Process Ontology from MSigDB" -> I do not understand what exactly is a subgraph in this setting. * Similarly, the authors could give an idea how the EM User graph looks like. Since it is no standard KG/domain used in the GNN area, it is hard to imagine how the subgraphs are connected if you do not know it. * The authors refer to the appendix for details about the datasets. I would suggest to add to the paper at least a few short notes what is specific/different for each. For example, for standard graphs, we have that KGs are rather sparse, biomedical graphs are rather dense, etc. Do the proposed datasets show and/or differ in such characteristics too? * Bibliography contains references to arxiv papers, while the papers actually have been published at official venues, e.g., [50], [51]


Review 3

Summary and Contributions: I acknowledge the rebuttal. The reviewers provided more information about efficiency in the paper. I raised my vote to accept. ========================================================== The authors proposed a subgraph neural network to learn disentangled subgraph representations. Three channels are designed to capture different aspects of subgraph topology. Experiments results on both synthetic and real-world datasets demonstrate the effectiveness of the proposed model in subgraph classification.

Strengths: 1) The authors studied a relevant problem and the proposed approach is novel and looks reasonable. 2) The experimental results in both synthetic and real-world datasets show the effectiveness of the proposed model. 3) The source code is provided to make the work reproducible.

Weaknesses: My only concern lies in efficiency. Although the authors mentioned precomputed similarities can speed up the optimization, subgraph similarity calculation is time-consuming. Also it would be interesting to show the speedup of the precomputing strategy empirically. Besides, it would be useful to report not only the computational complexity but also the running time.

Correctness: Yes

Clarity: Yes

Relation to Prior Work: Yes

Reproducibility: Yes

Additional Feedback:


Review 4

Summary and Contributions: The reviewers answered my main complaint: They have run the baseline I had in mind but had incorrectly labeled it in the paper. I am raising my vote to accept. ----------- Presents a graph neural network that accepts a graph, and a "query" (subset of the powerset of the graph), and returns one graph label for each subgraph in the query. The writing does not say it in this concise way above, does not consider simple baselines (run GCN on each subgraph), and the overall system has many components, functions and heuristics, making it an implementation paper, more than a general advancement.

Strengths: * Point-out a missing gap in literature (subgraph prediction); though trivial baselines not evaluated (see Weakness #2). * Apply on science datasets (biology and chemistry)

Weaknesses: * Why is baseline (7) "pretrained"? To me, that is the most straight-forward implementation (treat each graph like a subgraph) and run it through GNN (e.g. GCN of Kipf & Welling) * The function that actually does the random walk (biased random walk) is a heuristic. It needs to be separately tuned. * The writing is too verbose and hard to follow. It looks like an implementation paper and not-so-much a theoretical nor a general advancement. This is probably the biggest weakness IMO. I suggest you focus on a main idea or two and show analyses why they work, rather than a well-engineered system with many functions and things to be designed/considered.

Correctness: Equation 2 is hard to interpret because subscript "c" on the left is not used on the right. Typos: Line 42: "our region" --> "one region" ? Line 76: Grammar mistake Line 118: Add $\in \mathcal{S}$ before the period for exactness. Other typos: Line 131

Clarity: The paper is too verbose and the story-line is not smooth to follow. Math is often incomplete (why not specify domains and ranges for all functions?) and definitions are not exact.

Relation to Prior Work: Yes! I think they covered all relevant aspects and the related work section is very well written, thank you!

Reproducibility: No

Additional Feedback: Reproducibility: Since the model is very complicated, I dont think I am able to implement their model by reading their paper (For reference, I have well-cited ML-graph papers at ICML&NeurIPS) If other reviewers disagree: e.g. a student in their lab can implement and get the results, from this state of the writing, then I am willing to change my mind about this.

[Author Response · NeurIPS 2020]

We thank the reviewers for their time and valuable feedback that will strengthen our paper. Overall, we are glad the reviewers agree that our paper identifies general subgraph prediction as a sizable gap in the GNN literature, introduces a sound and interesting initial solution to fill this gap, and provides 7 baselines and 8 new datasets to jump-start this area of research. We think the following clarifications and new analyses resolve all key issues raised.

**(1) Novelty & general advancement.** R1 and R4 questioned our theoretical or general advancement. We respectfully disagree and would like to clarify our paper's five key advancements. **1)** SUB-GNN *"proposes a new field of study: subgraph classification"*, *"is the first method that works with varying subgraph sizes"*, and *"can easily deal with large subgraphs"* (R1). **2)** In our initial solution for subgraph classification, the taxonomization of subgraph topology (Table 1) is grounded in network science theory [Yang & Leskovec, ICDM 2012], and our design of disentangled channels is directly rooted in these theoretical properties. **3)** We propose 7 strong baselines, each carefully designed to test SUB-GNN and its components. **4)** We construct 8 new and original datasets. **5)** SUB-GNN is a flexible framework for learning subgraph embeddings—it can operate with any sampling scheme, patch embedder, and similarity function.

**(2) Related work & baselines. (2.1)** R1 pointed out a missing reference to Meng et al, 2018, which, unlike SUB-GNN, performs prediction on small (3-4 nodes), fixed-size subgraphs. We extensively studied this interesting paper, but left out the citation in error. **(2.2)** R4 requested a baseline in which a GNN is run on each subgraph. We agree this baseline is important, and it already appears in our paper (L275, Baseline 7). We believe the confusion occurred because we incorrectly stated that the model uses pretrained node embeddings (L275). In fact, the GNN model (GIN) trains node embeddings *from scratch* on each task. We will correct this typo in the final version.

**(3) Run-time.** R3 raised an important question about the efficiency of SUB-GNN and the performance gains from pre-computation. We find that pre-computation results in an 18X training speedup. With pre-computation, training SUB-GNN on PPI-BP takes 30s per epoch on a single GPU, of which 25%, 43%, and 32% are spent on neighborhood, position, and structure channels, respectively. We will include a full run-time analysis in the final version. We note that training time can be further improved via locality sensitive hashing or k-hop shortest paths for similarity calculations.

**(4) Reproducibility and code.** We thank R2 and R3 for pointing out that *"the source code is provided to make the work reproducible."* R4 and R1 questioned whether our descriptions are sufficient to implement the model. To address this, we included in our initial submission a link to the complete SUB-GNN implementation, together with examples of usage, baseline implementations, and hyperparameters. Furthermore, we appreciate the reviewers' suggestions for improving the clarity in our writing, and will incorporate them in the final version to improve reproducibility.

**(5) Datasets. (5.1)** R1 raised an important point on the possibility of information leakage in case train and test subgraphs overlapped. While we disagree that this constitutes information leakage, as subgraphs with overlapping nodes can have different labels, we strongly agree that assessing performance as a function of overlap in nodes between train and test subgraphs is important. We have two answers to this point. First, test subgraphs in the COMPONENT and CORENESS datasets have zero nodes in common with any train or validation subgraphs. Our strong performance on these datasets (Table 2) indicates that node overlap between subgraphs does not explain SUB-GNN generalization power. Second, we generated an extra COMPONENT dataset with varying degrees of overlap between the test and train/val subgraphs. We find that SUB-GNN performs strongly on both test subgraphs with low overlap (<10%) and test subgraphs with high overlap (>50%) (92.9 vs 96.1 F1-score). **(5.2)** R2 questioned whether the HPO tasks are *"actually done in practice."* HPO tasks are of incredible practical relevance. For example, the Undiagnosed Diseases Network, a major program backed by NIH, seeks to diagnose patients from HPO codes [Splinter et al. NEJM 2018]. We designed the HPO datasets to directly mimic this task of rare disease diagnosis [Bradley, Nature Reviews 2020; Austin & Dawkins, Nature 2017].

**(6) Further considerations. (6.1)** R1 questioned the effect of different node representations and similarity functions. In a new experiment, we used GraphSAINT [Zeng et al. ICLR 2020] to learn node representations. SUB-GNN outperforms the strongest baseline by 10.5% on average on synthetic datasets. We will experiment with graph edit distance as the similarity function for the structure channel and include in the final version. **(6.2)** R1 raised an important point about whether SUB-GNN can be inductive. SUB-GNN is inductive because it can generalize to new subgraphs on unseen portions of the graph, a notion similar to that defined in Meng et al. 2018. **(6.3)** R1 asked several questions about our anchor patch sampling procedure. We do not average over all possible anchor patches. For the structure and border position channels, anchor patches are sampled independently of the subgraphs and shared across subgraphs; this is broadly inspired by the shared anchor-set sampling in P-GNN [You et al, ICML 2019], and is inductive to new subgraphs. For the neighborhood channel, sampling is a standard k-hop neighborhood sampling procedure, performed locally within and around each subgraph. **(6.4)** R1 rightly pointed out that permutation invariance of $h_{X,c}$ does not remove all position information. We agree and will correct this in the final version. The order invariant representations are needed for layer-to-layer message passing. **(6.5)** R1 noted that SUB-GNN has many hyperparameters and raised the important question of how to set them without high family-wise error rates. We used random search (see Appendix and submitted code). Importantly, we performed the same number of hyperparameter searches on all baselines and SUB-GNN. Of the hyperparameters we optimized, most are standard in modern neural networks. Only 7 (border sizes, number of anchor patches, etc.) were peculiar to our model, some of them indicating flexibility of SUB-GNN, e.g., to turn on/off a channel. We observed similar performance on validation and test sets, indicating no overfitting.

[Meta-Review · NeurIPS 2020]

There was healthy discussion and updating of opinions on this paper. The consensus is strongly in favor of acceptance. However, this opinion is under the assumption that the authors will make changes promised in the rebuttal. Please go very carefully over the reviews and make sure that the promised changes are made, and that as much as possible is done to address the comments raised by the reviewers.